# Cells Lacking PA200 Adapt to Mitochondrial Dysfunction by Enhancing Glycolysis via Distinct Opa1 Processing

**DOI:** 10.3390/ijms22041629

**Published:** 2021-02-05

**Authors:** Abdennour Douida, Frank Batista, Pal Boto, Zsolt Regdon, Agnieszka Robaszkiewicz, Krisztina Tar

**Affiliations:** 1Department of Medical Chemistry, Faculty of Medicine, University of Debrecen, Egyetem tér 1, H-4032 Debrecen, Hungary; abdoallah461@yahoo.fr (A.D.); regdon.zsolt@med.unideb.hu (Z.R.); 2Doctoral School of Molecular Medicine, University of Debrecen, H-4032 Debrecen, Hungary; 3Department of Biochemistry and Molecular Biology, Faculty of Medicine, University of Debrecen, H-4032 Debrecen, Hungary; frank.batistapelaez@gmail.com; 4Stem Cell Differentiation Laboratory, Department of Biochemistry and Molecular Biology, University of Debrecen, H-4032 Debrecen, Hungary; boto.pal@med.unideb.hu; 5Department of General Biophysics, Faculty of Biology and Environmental Protection, University of Lodz, 90-236 Lodz, Poland; robaszkiewicz.agnieszkaz@gmail.com

**Keywords:** PA200, RNA-seq, metabolism, mitochondria, glycolysis, Opa1

## Abstract

The conserved Blm10/PA200 proteins are proteasome activators. Previously, we identified PA200-enriched regions in the genome of SH-SY5Y neuroblastoma cells by chromatin immunoprecipitation (ChIP) and ChIP-seq analysis. We also found that selective mitochondrial inhibitors induced PA200 redistribution in the genome. Collectively, our data indicated that PA200 regulates cellular homeostasis at the transcriptional level. In the present study, our aim is to investigate the impact of stable PA200 depletion (shPA200) on the overall transcriptome of SH-SY5Y cells. RNA-seq data analysis reveals that the genetic ablation of PA200 leads to overall changes in the transcriptional landscape of SH-SY5Y neuroblastoma cells. PA200 activates and represses genes regulating metabolic processes, such as the glycolysis and mitochondrial function. Using metabolic assays in live cells, we showed that stable knockdown of PA200 does not change basal respiration. Spare respiratory capacity and proton leak however are slightly, yet significantly, reduced in PA200-deficient cells by 99.834% and 84.147%, respectively, compared to control. Glycolysis and glycolytic capacity show a 42.186% and 26.104% increase in shPA200 cells, respectively, compared to control. These data suggest a shift from oxidative phosphorylation to glycolysis especially when cells are exposed to oligomycin-induced stress. Furthermore, we observed a preserved long and compact tubular mitochondrial morphology after inhibition of ATP synthase by oligomycin, which might be associated with the glycolytic change of shPA200 cells. The present study also demonstrates that the proteolytic cleavage of Opa1 is affected, and that the level of OMA1 is significantly reduced in shPA200 cells upon oligomycin-induced mitochondrial insult. Together, these findings suggest a role for PA200 in the regulation of metabolic changes in response to selective inhibition of ATP synthase in an in vitro cellular model.

## 1. Introduction

Glycolysis and oxidative phosphorylation (OXPHOS) are the major energy providers to cells [1,2]. In aerobic cells, the vast majority of chemical energy is produced by the mitochondria as ATP. The mitochondrial respiratory complexes (complexes I–IV) perform the multiple steps of oxidative phosphorylation; electrons pass through each complex to the final electron acceptor, oxygen, which is reduced to water. The transfer of electrons creates an electrochemical proton gradient across the inner mitochondrial membrane, which potentiates the production of ATP by ATP synthase [3,4]. ATP production matches cellular needs. Cells adapt to higher cellular energy demands during stress or metabolic adaptation by maximizing the mitochondrial respiration capacity to phosphorylate more ADP to ATP. Shifts between glycolysis and mitochondrial respiration are also observed when cells adapt to environmental changes or when mitochondria become dysfunctional [5,6,7]. Similar phenomena occur in many cancers. Even in the presence of oxygen, cancer cells exhibit increased glycolysis to generate rapid, accessible ATP. The metabolic shift from OXPHOS to aerobic glycolysis in tumors is associated with increased cell proliferation and cell survival [8,9,10].

Mitochondrial function is tightly controlled and linked to mitochondrial dynamics [11,12]. Emerging data suggest that alterations in mitochondrial dynamics strongly regulate mitochondrial activity. Mitochondrial dynamics may also modulate mitochondrial bioenergetics [13]. Many diseases, including neurodegenerative diseases, cardiovascular diseases, and many cancers exhibit perturbations in mitochondrial physiology [14,15,16,17,18]. Regulation of the mitochondrial network and the fusion–fission machinery may modulate mitochondrial energetics to initiate crosstalk between signaling components and the cell-cycle machinery [19,20].

Mitochondria continuously fuse and divide, and the morphology of the organelle is mainly determined by specific GTPases. Mitochondrial fission is predominately regulated by the GTPase activity of Drp1 (dynamin-related protein-1), a cytosolic protein. Mitochondrial fission provides quality control for the organelle [21,22,23]. Drp1 and its yeast ortholog Dnm1 are large dynamin-related GTPases that cooperate with adaptor proteins to promote mitochondrial fission. Dnm1/Drp1 proteins assemble into oligomeric structures [24,25,26,27]. They are mechanochemical enzymes that use their GTPase activity to drive membrane fission [28]. Opa1 (optic atrophy protein-1), on the other hand, is found in the mitochondrial inner membrane and is involved in inner-membrane fusion and protection from apoptosis [29,30,31]. Furthermore, Opa1 plays a role in mitochondrial cristae remodeling independent of mitochondrial fusion [32,33]. Depletion or mutation of Opa1 leads to slowed cellular growth and a deficit in mitochondrial respiration [34].

Recently we have identified PA200-enriched regions in the genome of the human neuroblastoma cell line SH-SY5Y using chromatin immunoprecipitation (ChIP) and ChIP-seq analysis [35]. PA200 and its yeast ortholog Blm10 are monomeric proteins that activate the proteasome and facilitate the degradation of unstructured proteins independent of ubiquitin and ATP [36,37,38,39,40,41]. When cells were treated with selective mitochondrial toxins, PA200 was redistributed in the genome. Genes whose promoters were enriched regulate cell proliferation, modifications of proteins, and cell metabolism. Thus, in the present study, our aim was to perform RNA-seq and RNA-seq analysis in cells stably depleted of PA200 and control. We demonstrated that depletion of PA200 contributes to overall changes in the transcriptional landscape in neuroblastoma cells. PA200 activates and represses genes involved in MAPK and JNK signaling and genes regulating metabolic processes, such as glycolysis and mitochondrial function. We show that the genetic ablation of PA200 induced mitochondrial dysfunction and increased glycolysis, suggesting a shift from oxidative phosphorylation to glycolysis, especially when cells are exposed to stress. The present study also demonstrates that OMA-1-dependent Opa1 processing is affected in PA200-deficient cells, suggesting a role for PA200 in the regulation of metabolic changes in response to stress.

## 2. Results

### 2.1. PA200 Regulates Transcription of Genes Involved in the Key Cellular Processes

Our previously published data demonstrated the occurrence of PA200 at the promoters of genes involved in crucial intracellular processes, including survival, proliferation, protein modifications, and metabolism in SH-SY5Y [35]. Thus, we tested if PA200 is a bona fide regulator of gene transcription. For this reason, we made use of the stably generated SH-SY5Y cell line with PA200 knockdown (Appendix A) and compared the transcriptome of shPA200 (stably depleted of PA200) cells and the corresponding control-shCTRL cells using mRNA sequencing. The analysis of differential gene expression between the two cell lines confirmed the substantial reduction of *PSME4* mRNA (gene name for PA200) in the SH-SY5Y knockdown cells (Figure 1A).

PA200 deficiency caused considerable alterations in gene expression (Figure 1B,C and Appendix A). In the relatively long list of differentially expressed genes (DEGs), PA200 emerged as both a transcription activator and repressor (Figure 1D,E). The functional annotation of PA200-up-regulated and down-regulated genes uncovered processes that are crucial for proper cell functioning and response to external stimuli (Figure 1D,E and Appendix A). These processes include, but are not limited to, neuron differentiation, signal transduction, the MAPK signaling pathway, histone modification, DNA repair and proliferation, cell-cycle regulation, cellular response to oxidative stress, and apoptotic processes and cell death. These results are in accordance with our previously published data, in which the regulation of many genes was confirmed by quantitative real-time PCR (35). The greatest representation of DEGs was assigned to cellular metabolism (Figure 1D,E and Figure 2A).

PA200 was found to affect ubiquitin-dependent and proteasomal-protein catabolic processes, ATP production, and reducing-power (NAD and NADP) regeneration. The stable silencing of PA200 also resulted in the modified expression of genes involved in inter alia glycolysis and oxidative phosphorylation (Figure 1D,E and Figure 2B) processes that determine the energetic and metabolic state of cells.

### 2.2. Mitochondrial Stress Assay Indicates Mitochondrial Dysfunction in shPA200 Cells

We have shown that the deletion of BLM10, the yeast orthologue of PA200, resulted in dysfunctional mitochondria with reduced respiratory capacity and reduced fitness during respiratory growth [42]. Thus, in search of further evidence that PA200 might modify cellular metabolic processes, we investigated the mitochondrial bioenergetics profile using the cell stress mito test by Seahorse analysis following the mitochondrial metabolic insult. We measured oxygen consumption rate (OCR) in real-time in a cell line with stably depleted PA200 (shPA200) and its corresponding control. We measured OCR at the basal level and OCRs after adding selective mitochondrial inhibitors in sequential order.

We used oligomycin, cyanide-p-trifluoromethoxyphenylhydrazone (FCCP), and a combination of rotenone and antimycin A. Oligomycin binds and inhibits ATP synthase to prevent protons from passing back into the mitochondria. FCCP uncouples mitochondrial respiration and reduces the synthesis of ATP by collapsing the proton gradient across the mitochondrial inner membrane. Therefore, we could measure uncoupled mitochondrial respiration (maximal respiration). The complex I inhibitor rotenone and the complex III inhibitor antimycin A shut down mitochondrial respiration, so mitochondrial and nonmitochondrial oxygen consumption could be distinguished. Spare respiratory capacity is regarded as the reserved capacity of a cell to generate ATP by oxidative phosphorylation when energy is in demand [4,43].

Following normalization to total cellular protein levels, our analysis indicated a significantly reduced proton leak (84.147% reduction compared to control) and reduced reserved (spare) respiratory capacity (99.834% reduction compared to control) in shPA200 cells compared to control cells (Figure 3A,B). These results suggest that the extra ATP production by oxidative phosphorylation during energy demand is impaired in cells depleted of PA200. Maximal respiration following FCCP administration was also decreased in shPA200 compared to control cells. Although the effect was not significant (*p* = 0.07), this result suggests mitochondrial dysfunction.

To determine whether the changes in mitochondrial metabolic profiling alter the levels of core proteins of the OXPHOS machinery, we used total cell lysates (Figure 3C and Appendix A) and purified mitochondria (Figure 3E and Appendix A). Loss of PA200 led to a significantly reduced protein level of the accessory subunit NDUFB8, a member of complex I. It is required for the assembly of the functional complex, but may not be involved in the catalytic activity of complex I (Figure 3D,F) [44].

Subsequently, we performed mitochondrial-membrane-potential measurements. To detect mitochondrial membrane potential by flow cytometry, we used TMRE following 24 h treatment with DMSO and 3 µM oligomycin. When cells were treated with 3 µM oligomycin, the mitochondrial membrane potential was significantly increased in both control and shPA200 cell compared to vehicle-treated control. In summary, the loss of PA200 slightly potentiated the effect of oligomycin, compared to control cells, causing increased hyperpolarization of mitochondrial membrane potential. The effect however did not reach significance (Appendix A). We compared intracellular ROS in shPA200 and control cells using carboxy-H2DCFDA with and without oligomycin treatment. We found significantly elevated cytosolic ROS in oligomycin-treated shPA200 cells compared to vehicle-treated shPA200 cells, which indicates existing oxidative stress in our cell model (Appendix A).

### 2.3. The Genetic Ablation of PA200 Results in Increased Glycolysis and Glycolytic Capacity

We next sought to determine if the metabolic stress response induced changes in glycolysis after PA200 depletion. For this purpose, we performed the glycolysis stress test in real time on intact shPA200 and control cells by measuring extracellular acidification rate (ECAR) using Seahorse XF analysis. We sequentially added glucose, oligomycin, and 2-deoxyglucose (2-DG) to our samples. Glucose was added to measure and calculate glycolysis under basal conditions, based on the differences in values of ECAR before and after glucose administration. Oligomycin administration was added to determine the glycolytic capacity of cells: oligomycin blocks oxidative phosphorylation and, thus, directs cells toward glycolysis to fulfill ATP demand. The glucose analog, 2-DG, inhibits glycolysis to determine baseline ECAR values. Interestingly, the glycolytic activity was significantly higher (42.186% increase) in shPA200 cells compared to control cells. Furthermore, energy production (glycolytic capacity), independent of mitochondrial respiration was also slightly (26.106%), yet significantly, higher, suggesting a metabolic shift from oxidative phosphorylation to glycolysis in cells depleted of PA200 (Figure 4A,B).

### 2.4. Preserved Long and Compact Tubular Mitochondrial Morphology in shPA200 Cells after Selective Mitochondrial Inhibitor Treatment

Mitochondrial dynamics contribute to mitochondrial function [45]. Thus, we explored whether selective mitochondrial inhibitor treatment would result in different morphological changes in mitochondria in shPA200 cells compared to the control. We challenged cells with oligomycin, a selective mitochondrial ATPase synthase inhibitor. Oligomycin induces mitochondrial fragmentation [46]. Furthermore, we recently showed that loss of PA200 does not sensitize cells to death after 3 µM oligomycin treatment, in contrast to rotenone treatment. We also demonstrated that PA200 depletion causes cells to accumulate in the S phase of the cell cycle after treatment with oligomycin, indicative of possible delay of DNA replication [35]. Oligomycin was described as an oncogenic agent in several types of lung cancer cells, causing increased cell invasion and migration [47]. Oligomycin was validated as a relevant tool to study bioenergetics adaptation to OXPHOS suppression in many cancer-cell lines [48]. Moreover, our ChIP-seq data showed PA200-enriched regions in the genome of SH-SY5Y and that the status of binding or eviction of PA200 to/from specific promoters depends on the exposure to selective mitochondrial toxins including oligomycin. GO annotation revealed that many genes that were significantly enriched in PA200 contribute to the regulation of metabolism [35].

Thus, as a logical step, we also determined the effects of PA200 deficiency on mitochondrial morphology after oligomycin treatment. Using live-cell high-content-imaging analysis with Mitotracker Red CMXRos to stain the mitochondria, we quantified mitochondrial species in the presence or absence of 3 µM oligomycin (Figure 5A–C). We classified mitochondria as long tubular, short tubular, compact tubular, and fragmented (Figure 5D) using Harmony 4.8 and PhenoLogic machine-learning software.

We observed no significant differences between vehicle-treated shPA200 and vehicle-treated control cells. In both cell lines, elongated mitochondrial morphology was maintained (Figure 6A,B).

Treating cells with 3 µM oligomycin resulted in increased fragmentation of mitochondria in both cell lines, but the effect was significantly less pronounced in PA200 knockdown cells. Moreover, long and compact tubular mitochondrial structures were significantly higher in cells depleted of PA200 compared to control (Figure 6C left panels) following oligomycin treatment. In summary, the increased glycolysis and glycolytic capacity of shPA200 were observed in association with preserved long and compact tubular mitochondrial morphology after mitochondrial stress.

### 2.5. Effects of PA200 Knockdown on mRNA Expression of Genes Related to Mitochondrial Fusion and Fission

The mitochondrial fusion–fission process governs the maintenance of mitochondrial morphology [32,49,50,51]. We, therefore, analyzed the mRNA levels of the mitochondrial fusion protein genes, *OPA1* (Opa1), *MFN1* (Mitofusin-1), and *MFN2* (Mitofusin-2), and the mitochondrial fission protein genes, *MIEF1* (Mid51), *MIEF2* (Mid49), *FIS1* (Fis1), *DNM1L* (Drp1), and *MFF* (Mff), in control and shPA200 cells treated with vehicle (DMSO) and 3 µM oligomycin (Figure 7A). Figure 7A demonstrates gene-expression fold changes following 24 h treatment with vehicle and 3 μM oligomycin.

The expression fold changes in shPA200 cells were compared to the control cell line with the respective treatment (DMSO or 3 μM oligomycin). As shown in Figure 7A, no significant changes due to PA200 depletion were detected compared to the respective controls, except a slight but significant increase in the fold change of *MFN2* (Mitofusin-2) in the vehicle-treated shPA200. We also analyzed the mRNA changes in both cell lines in response to the 3 µM oligomycin treatment. The values were normalized to the corresponding vehicle-treated cell line. The data indicate a significant reduction of mRNA expression of *MIEF2* (MiD49) in both cell lines, and a significant decreased *MIEF1* (MiD51) in shPA200 cells. MiD51 and MiD49 are two recently discovered components of mitochondrial fission that govern the recruitment of Drp1 to the mitochondrial surface. Knockdown of these two genes results in fusion [52,53]. The mRNA expression levels of *MFN1*, *MFF,* and *FIS1* were significantly reduced in both cell lines. In summary, oligomycin modifies the gene-expression level of mitochondrial fusion–fission proteins in both control and shPA200 cells. However, the depletion of PA200 did not cause significant changes in the mRNA expressions of the mitochondrial fission–fusion machinery compared to the control cells, except for *MFN1* after oligomycin treatment (Figure 7B) and *MFN2* after treatment with vehicle (Figure 7A).

### 2.6. Genetic Ablation of PA200 Leads to Changes in Opa1 Processing in Cells Exposed to Selective Mitochondrial Insult

According to the quantitative real-time PCR results, no major changes in gene expression of mitochondrial fission–fusion proteins occur in response to PA200 depletion. In addition, oligomycin treatment induced similar changes in mitochondrial fission–fusion genes in both control and shPA200 cells. However, we still demonstrated inefficient mitochondrial function, significant differences in mitochondrial morphology, and elevated glycolytic capacity after selective mitochondrial inhibition in shPA200 cells. A recent report indicated that enhanced glycolysis in mesenchymal cells promoted cell survival via Opa1-mediated fusion, regulated by leptin [54]. Another study also demonstrated the involvement of Opa1 in glycolytic ATP production for microtubule network assembly and neutrophil extracellular trap formation [55]. Thus, we hypothesized that the significantly higher percentage of mitochondria with long and compact tubular structures and the elevated glycolytic capacity in shPA200 cells after mitochondrial stress might originate from changes in Opa1 processing. Mitochondrial dysfunction with reduced ATP levels may stimulate Opa1 processing, resulting in mitochondrial fragmentation [56,57]. Oligomycin treatment can activate the proteolysis of L-Opa1 [58]. Opa1 has many isoforms resulting from alternative splicing and proteolytic processing. Proteolytic cleavage of Opa1 occurs at two sites, S1 and S2, governed by different enzymes. The delicate balance of long (L) and short (S) isoforms of Opa1 is requisite for mitochondrial fusion. Low mitochondrial ATP, apoptotic stimuli, and dissipation of mitochondrial membrane potential promote Opa1 processing, leading to the accumulation of short isoforms and the loss of long isoforms, which promote mitochondrial fission [56,57,59]. Thus, the level of mitochondrial fission and fusion proteins (Figure 8 and Appendix A) was also assessed by western blot. No major changes in mitochondrial fission and fusion proteins in shPA200 cells were detected compared to control after vehicle treatment (Figure 8A). Treating the cells with 3 µM oligomycin resulted in a major reduction of Fis1 and MFN1 in both cell lines (Figure 7B). Inhibition of the F_1_F_0_-ATP synthase by 3 μM oligomycin treatment did not reduce MFN2 levels in any cell lines (Figure 8A). However, the oligomycin treatment significantly reduced the level of L-Opa1 (upper band) and led to the accumulation of S-Opa1 (lower band) in control cells, in accordance with a previous study [60] (Figure 8B). Interestingly, the oligomycin-induced L-Opa1 cleavage was attenuated in cells stably depleted of PA200 (Figure 8B).

Opa1 processing is initiated by the IM peptidase, OMA1, and the *i*-AAA protease, YME1L [61,62]. These two enzymes cleave Opa1 at S1 and S2, respectively. OMA1 is required for stress-induced Opa1 cleavage and mitochondrial fragmentation. However, depletion of YME1L impairs the constitutive processing of Opa1 at S2, leading to mitochondrial fragmentation that makes cells susceptible to apoptosis [63].

Next, we determined if the preserved L-Opa1 levels in shPA200 upon oligomycin treatment originates from changes in proteins that regulate Opa1 processing. Figure 8C demonstrates that the proteolytically active OMA1 level significantly decreased in shPA200 after oligomycin treatment. This result suggests that stress-induced cleavage of Opa1 and thus fragmentation of mitochondria is reduced. The level of YME1L did not change after treatments, suggesting that constitutive processing of Opa1 by YME1L was maintained (Figure 8D).

## 3. Discussion

We have previously identified PA200-enriched regions in the genome of the human neuroblastoma cell line SH-SY5Y. Binding/eviction of PA200 to/from promoter regions was dependent on selective mitochondrial inhibitors. We have found that the proteasome activator PA200 is recruited to the chromatin and is associated with promoters of genes involved in the cell cycle, primary metabolism, and protein modification processes. We identified and validated genes regulating apoptosis, cell proliferation, and survival, whose expressions are influenced by PA200 [35]. In this study, we performed RNA-sequencing and overall transcriptomic analysis, using the human neuroblastoma cell line stably depleted of PA200 and its respective control to evaluate gene-expression changes. Global transcriptome profiling revealed that expression of genes was both repressed and activated by PA200. A recent report by JiangTian-Xia et al. [64] demonstrated that PA200 maintains the stability of histone marks during transcription and aging. Using mouse embryonic fibroblasts (MEF) and mouse liver from a PA200 knockout mouse, they performed RNA-seq. Similar to our previous results, they suggest that the deletion of PA200 influences transcriptomic regulation, including gene-expression changes in the cell cycle and MAPK signaling.

Following bioinformatics analysis of our RNA-seq metadata, we noticed that many genes that are up- or down-regulated upon stable depletion of PA200 are related to or participate in metabolic processes, including glycolysis, ATP, and NADH generation, and mitochondrial homeostasis. Yeast cells with the deficiency of *BLM10*, the yeast orthologue of PA200, exhibited a dysfunctional mitochondrial phenotype [37,42]. The loss of *BLM10* resulted in reduced respiratory capacity and increased oxidative damage to the mitochondria. Yeast cells depleted of *BLM10* were unable to grow on nonfermentable carbon sources, such as glycerol. Moreover, the expression of *BLM10* was strongly induced when cells were switched from fermentation to oxidative metabolism [65]. These observations in yeast and the results of DEG clustering of the RNA sequencing data directed us to investigate the metabolic state of cells stably depleted of PA200.

In this study, we measured mitochondrial respiration (OCR) and glycolytic flux (ECAR) in control and shPA200 human neuroblastoma cells. In our study, the spare respiratory capacity of shPA200 cells was severely impaired, indicating that the ability of cells to provide energy by oxidative phosphorylation in a case of a sudden demand for ATP was affected. Cells need more energy to maintain cell function when exposed to stress. Increased spare respiratory capacity is a good indicator of cells’ ability to provide more energy by oxidative phosphorylation to overcome stress [4,43,66]. Our data demonstrate that the loss of PA200 inhibited mitochondrial function; the coping mechanism, which provides more energy by oxidative phosphorylation during increased cellular demand, was compromised. On the other hand, according to the glycolytic stress assay, glycolysis and glycolytic capacity increased significantly in shPA200 cells, when cells were further exposed to mitochondrial crisis. These data suggest that an inefficient mitochondrial function in cells stably lacking PA200 may drive cells to increase glycolysis, producing more ATP to overcome mitochondrial crisis-induced stress. Our data collectively indicate that cells lacking PA200 exhibit inefficient mitochondrial function: cells can maintain cell function if they are not exposed to further stress with increased energy demand. However, when shPA200 cells were exposed to selective mitochondrial insult (blocking the F_1_F_0_-ATP synthase with oligomycin), the cells adapted by switching from oxidative phosphorylation to increased glycolysis to provide ATP. These results are consistent with our previously published data demonstrating that shPA200 cells were not sensitized to oligomycin-induced cell death.

There is a tight link between mitochondrial function, changes in metabolic regulation, and mitochondrial morphology. Balanced mitochondrial fusion and fission processes govern mitochondrial dynamics. Emerging data provide evidence that mitochondrial dynamics are involved in many pathophysiological processes [67]. Key mitochondrial fission and fusion proteins, including Drp1 (fission), Mitofusin 1 and 2, and Opa1 (fusion), orchestrate this process. Changes in the dynamic nature of the mitochondrial network affect mitochondrial bioenergetics and vice versa. A recent publication demonstrated that mitochondrial bioenergetics are regulated by the circadian oscillation of Drp1 activity and that mitochondrial dynamics are clock-controlled via Drp1 activity regulation [68]. Furthermore, highly fused mitochondria are regulated mainly by Opa1, and fusion is positively associated with increased ATP production, while fragmented mitochondria help to maintain the quality control of the organelle [69].

As a logical step, we investigated the mitochondrial morphology in control and shPA200 cells using high content screening provided by Opera Phenix. High xontent screening provided us with high-quality confocal images and enabled us to perform advanced analyses [46]. Our results demonstrate that shPA200 cells do not exhibit major mitochondrial morphology differences under normal conditions. However, when we challenged cells with selective mitochondrial stress and blocked ATP synthesis, mitochondria organization shifted to long and compact tubular structures and less fragmentation in shPA200 cells compared to control cells. These results indicate either reduced fragmentation or induced fusion of the mitochondria network. Furthermore, L-Opa1 was preserved in cells depleted of PA200 following oligomycin treatment. These results combined with the increased glycolysis correlate with reduced levels of OMA1, the peptidase that promotes stress-induced cleavage of L-Opa1 to S-Opa1 leading to mitochondrial fragmentation.

In conclusion, PA200 appears to play a key role in metabolic adaptation and quality control under selective stress. PA200 regulates the expression of genes that are essential for diverse metabolic pathways. In the absence of PA200, cells exhibit mitochondrial dysfunction accompanied by an extensively reduced spare respiratory capacity. Decreased spare respiratory capacity indicates that cells are not able to provide enough energy by oxidative phosphorylation when stressed. Cells lacking PA200 adapt to mitochondrial dysfunction by switching from oxidative phosphorylation to glycolysis to provide energy and resist apoptosis. Our findings also indicate that the enhanced glycolysis might originate from reduced, stress-induced mitochondrial fragmentation through distinct Opa1 processing via decreased levels of OMA1. In summary, PA200 might be required for quality control of cells, because cells lacking PA200 show an alternative pathway to provide energy upon stress and, thus, can escape from cell death. However, the exact molecular mechanism requires further exploration.

## 4. Materials and Methods

All materials were purchased from Sigma-Aldrich (St. Louis, MO, USA) unless otherwise specified.

### 4.1. Cell Culture

Human SH-SY5Y neuroblastoma cells (European Collection of Authenticated Cell Cultures) were maintained in DMEM-high glucose supplemented with 10% heat-inactivated fetal bovine serum (FBS) (Gibco, ThermoFisher, Waltham, MA, USA), 2 mM L-glutamine, 100 units/mL penicillin, and 100 µg/mL streptomycin at 5% CO_2_ and 37 °C. Lentiviral technology was used to down-regulate the expression of PSME4/PA200 in the SH-SY5Y neuroblastoma cell line (shPA200) and to generate the appropriate control cell line (shCTRL) as described previously [35].

### 4.2. Mitotracker Red CMXRos Staining for High-Content-Screening Confocal Microscopy (HCS)

To stain the mitochondria, cells were seeded in cell carrier-96 ultra microplates (Perkin Elmer, Waltham, MA, USA) at a density of 1.5 × 10^4^ cell/well in complete DMEM with high glucose. Cells were treated for 24 h with vehicle (DMSO, 1% *v*/*v*) or 3 μM oligomycin. After 24 h, cells were rinsed with serum-free medium and fluoroBrit DMEM medium (Gibco, Thermo Fisher, Waltham, MA, USA), respectively. Cells were incubated at 37 °C in a 5% CO_2_ incubator with 50 nM Mitotracker Red CMXRos (Thermo Fisher, Waltham, MA, USA) and 10 μM Hoechst 33,342 (Thermo Fisher, Waltham, MA, USA) in serum-free medium for 30 min. Cells were washed twice with fluoroBrit DMEM medium and live cells were subjected to confocal imaging using an Opera Phenix high-content-screening system (Perkin Elmer, Waltham, MA, USA). Images were acquired while cells were incubated in 5% CO_2_ at 37 °C.

### 4.3. Mitochondrial Morphology Analysis

Automated confocal microscopy was performed on an Opera Phenix high-content-screening system (Perkin Elmer, Waltham, MA, USA). Image-acquisition settings were the following: 63× water objective (NA = 1.15), appropriate lasers and filters for Hoechst, eGFP, and Mitotracker Red in sequential mode to exclude spectra overlap. Detection was done with a 16-bit camera under nonsaturating conditions. Quantitative image analysis was performed with the built-in software (Harmony 4.8, Perkin Elmer, Waltham, MA, USA). Cell segmentation was performed based on Hoechst and EGFP staining to detect the nuclei and cytoplasm, respectively. Mitochondria were determined by the Mitotracker Orange signal using the Find-Spots building block. Spot properties were calculated (intensity, morphology, and texture) and mitochondria were classified based on measured properties using the PhenoLOGIC machine learning, as follows: long tubular, short tubular, compact tubular, and fragmented. The percentage of total mitochondria for each class was calculated using the following formula: (class area/total mitochondria area) * 100.

### 4.4. RNA Extraction and Quantitative RT-PCR

Total RNA was extracted using TRI Reagent (Molecular Research Center, Inc., Cincinnati, OH, USA) following the manufacturer’s protocol. Samples were treated with DNase I for 15 min at room temperature in DNA digestion buffer before the reverse transcription (Zymo Research, Irvine, CA, USA). To perform cDNA synthesis, a high-capacity cDNA reverse-transcription kit (Applied Biosystems, Foster City, CA, USA) was used to reverse transcribe 1 µg of total RNA with random primers. The quality of cDNA was checked by loading 1 µL of sample onto a 1% agarose gel.

### 4.5. Quantitative Real-Time PCR

Real-time PCR was performed with a LightCycler 480 Thermocycler (Roche, Basel, Switzerland) using SYBR Premix Ex Taq II (Takara Bio. Clontech Laboratories, Inc. Mountain View, CA, USA) according to the manufacturer’s protocol. Cycling conditions were as follows: Stage 1—initial denaturation 95 °C for 30 s, 1 cycle; Stage 2—PCR 95 °C for 5 s and 60 °C for 30 s, 40 cycles; and Stage 3—melt curve analysis 95 °C for 0 s, 65 °C for 15 s and 95 °C for 0 s, cooling 50 °C for 30 s, 1 cycle. Threshold values (Ct values) for all replicates were normalized to GAPDH and/or actin. Three technical replicates were measured for each biological replicate. To compare the effects of PA200 depletion and the effects of various treatments, 2^−ΔΔ*Ct*^ values were calculated to obtain fold expression levels [70]. The primer list is provided in Table 1.

### 4.6. Mitochondrial Fractionation

Mitochondrial fractionation of neuroblastoma cells was performed using a cell fractionation kit (Abcam, Discovery Drive Cambridge Biomedical Campus, Cambridge, UK) according to the manufacturer’s protocol.

### 4.7. SDS-PAGE and Western Blot

The shPA200 and control cells were rinsed with 1× PBS and lysed in RIPA buffer (50 mM Tris-HCl, 150 mM NaCl, 0.5% Na-deoxycholate, 2 mM EDTA, 1% NP-40, and 50 mM NaF) supplemented with protease-inhibitor cocktail (1 mM PMSF, 1 mM benzamidine and 1× EDTA-free protease-inhibitor cocktail) (cOmplete tablets Mini-EDTA-free, Roche, Basel, Switzerland). Cells were centrifuged at 12,000 rpm at 4 °C for 25 min. The supernatants were collected, and the protein concentration was estimated using Bradford assay (Quick Start^TM^ Bradford, Bio-Rad Laboratory, Hercules, CA, USA). Proteins (30 µg) from the total lysis were loaded and separated in 10% or 12% SDS-polyacrylamide gels. Proteins were transferred onto nitrocellulose membranes (0.45 µm NC Amersham, GE Healthcare Life Sciences, Chicago, IL, USA). For the mitochondrial respiratory chain-protein analysis (Total OXPHOS), an equal amount of protein from total lysate and the mitochondrial fraction from shPA200 and control cells were separated by western blot using the CAPS/PVDF transfer protocol. The membranes were blocked for 1 h with 5% (*w*/*v*) nonfat dry milk in 1× TBS-Tween20 (25 mM Tris-HCl, 150 mM NaCl, 0.05% Tween-20 pH 7.4) and then incubated overnight at 4 °C with primary antibodies. The bands were visualized using western-blotting luminol reagent (Santa Cruz Biotechnology, Dallas, TX, USA) after probing the primary antibodies using peroxidase-conjugated antibodies. β-actin or Hsp60 were used as a loading control. Images were taken by a ChemiDoc imager and signal intensity was analyzed using Image Lab V 6.1 software. The list of antibodies is provided in Table 2.

### 4.8. Seahorse XF 96 Flux Analysis

#### 4.8.1. Mitochondrial Stress Test Assay

The shPA200 or respective control cells were seeded (3.5 × 10^4^ cell/well) into an XF-96 cell-culture microplate (Seahorse Bioscience, Agilent, Chicopee, MA, USA) with the appropriate background correction wells. The cells were incubated overnight at 37 °C in a 5% CO_2_ incubator. DMEM high-glucose media supplemented with 10% FBS, 1× penicillin/streptomycin, 2 mM L-glutamine, and 1× anti-mycotic were used. In parallel, the sensor cartridge was prepared by adding 200 µL of Seahorse bioscience XF-96 calibrant solution (pH 7.4) (Seahorse Bioscience, Agilent, Chicopee, MA, USA) to each well of a 96-well utility plate. The sensors with the calibrant solution were incubated overnight at 37 °C without CO_2_. To measure the oxygen consumption rate (OCR), the growth medium was replaced by 180 µL XF assay medium supplemented with 4.5 g/L glucose and incubated at 37 °C in a CO_2_-free humidified incubator for 60 min. After calibration of the sensors, the basal OCR was determined 5 times for 30 min. The mitochondrial inhibitors used were sequentially injected at the following final concentrations: 1.5 µM oligomycin (Olig), 1 µM carbonyl cyanide-4 (trifluoromethoxy) phenyl hydrazone (FCCP), and 1 µM antimycin-A/rotenone (Anti/Rot). Measurements of OCR took place five times (5 min each) after each phase of drug injection. The OCR reads were normalized to total protein amount in each well. The protein concentration was measured using a quick start Bradford protein assay (Hercules, CA, USA). Data analysis was performed using Wave 2.3 Agilent Seahorse desktop software. Statistical analyses were assessed using Graphpad Prism 8.2.1 software.

#### 4.8.2. Glycolysis Stress Test Assay

The shPA200 or control cells were seeded (3.5 × 10^4^ cell/well) into an XF-96 cell-culture microplate (Seahorse Bioscience, Agilent, Chicopee, MA, USA). Cells were incubated overnight at 37 °C in a 5% CO_2_ incubator. The next day, the growth medium was replaced by 180 µL XF glucose-free assay medium (Seahorse Bioscience, Agilent, Chicopee, MA, USA) and incubated at 37 °C in a CO_2_-free, humidified incubator for 60 min. In parallel, the pre-incubated (overnight) XF-96 sensor cartridge was loaded for calibration (20 min). The extracellular acidification rate (ECAR) baseline was determined five times (5 min each), and then the drugs were sequentially injected at the following final concentrations: 10 mM glucose (Glu), 1 µM oligomycin (Olig), and 50 mM 2-deoxy-glucose (2-DG). The measurement of ECAR took place five times (5 min each) in each phase of injection. The ECAR data were normalized to total protein amount in each well. The protein concentration was measured as described above. Data analysis was performed using Wave 2.3 Agilent Seahorse desktop software. Statistical analyses were assessed using Graphpad Prism 8.2.1 software.

### 4.9. RNA-Seq

The shPA200 and control cells were cultured in T-75 flasks in DMEM high-glucose medium until 90% confluence. Total RNA was extracted according to the protocol described above (RNA extraction). Three independent experiments from each clone (control and shPA200) were performed. To obtain global transcriptome data, high throughput mRNA sequencing analysis was performed on the Illumina sequencing platform (Center for Clinical Genomics and Personalized Medicine, Core Facility, University of Debrecen, Debrecen, Hungary).

Total RNA sample quality was checked using an Agilent BioAnalyzer and a eukaryotic total RNA nano kit (Agilent, Chicopee, MA, USA) according to the manufacturer’s protocol. Samples with RNA integrity number (RIN) value >7 were accepted for the library preparation process.

RNA-Seq libraries were prepared from total RNA using a Ultra II RNA sample prep kit (New England BioLabs, Ipswich, MA, USA) according to the manufacturer’s protocol. Briefly, poly-A RNAs were captured by oligo-dT-conjugated magnetic beads, and then the mRNAs were eluted and fragmented at 94 °C. First-strand cDNA was generated by random-priming reverse transcription, and after second strand synthesis, double-stranded cDNA was generated. After repairing ends, A-tailing, and adapter ligation steps, adapter-ligated fragments were amplified in an enrichment PCR. Finally, sequencing libraries were generated. Sequencing runs were executed on Illumina NextSeq500 instrument using single-end 75-cycle sequencing.

### 4.10. RNA-Seq Data Analysis

Raw sequencing data (fastq) was aligned to the human reference genome version GRCh38 using the HISAT2 algorithm, and BAM files were generated. Downstream analysis was performed using StrandNGS software (www.strand-ngs.com (accessed on 1 February 2021)). BAM files were imported into the software, and the DESeq1 algorithm was used for normalization. To identify differentially expressed genes between conditions, a moderated *t*-test with Benjamini–Hochberg FDR for multiple testing correction was used.

### 4.11. Functional Analysis of RNA-Seq Data

The threshold for significantly up-regulated genes was set at 1.3 and down-regulated at 1.3 of the fold change in gene transcription between shPA200 versus shCTRL. The genes above and below the threshold values were considered differentially expressed genes (DEGs).

The heat map of the Log2 fold change (Log2FC) of gene transcription in shPA200 versus shCTRL was generated in the Heatmapper using average linkage for clustering and Euclidean distance measurement among differentially expressed genes. The statistical over-representation test for gene ontologies of biological processes was carried out in Panther, using Fisher’s exact test with no correction. Gene-regulatory signaling networks were generated in NetworkAnalyst. The number of nodes and edges were reduced using minimum network tool, and their colocation was set up using reduce overlap layout. Nodes representing particular processes (metabolism, MAPK cascade, PI3K-Akt signaling pathway, and the regulation of programmed cell death) were assigned to biological processes (Database:GO:BP) in Functional Explorer and highlighted in blue.

### 4.12. Statistical Analysis

Data from each experiment are summarized with the mean and standard deviation (SD) of *n* ≥ 3 biologically independent experiments. Statistical analyses were performed using ANOVA or unpaired student’s t-tests. GraphPad Prism V8.2.1 was used for statistical analyses, and the significance was determined as * *p* < 0.05, ** *p* < 0.01, *** *p* < 0.001, and **** *p* < 0.0001.

## Figures and Tables

**Figure 1 ijms-22-01629-f001:**
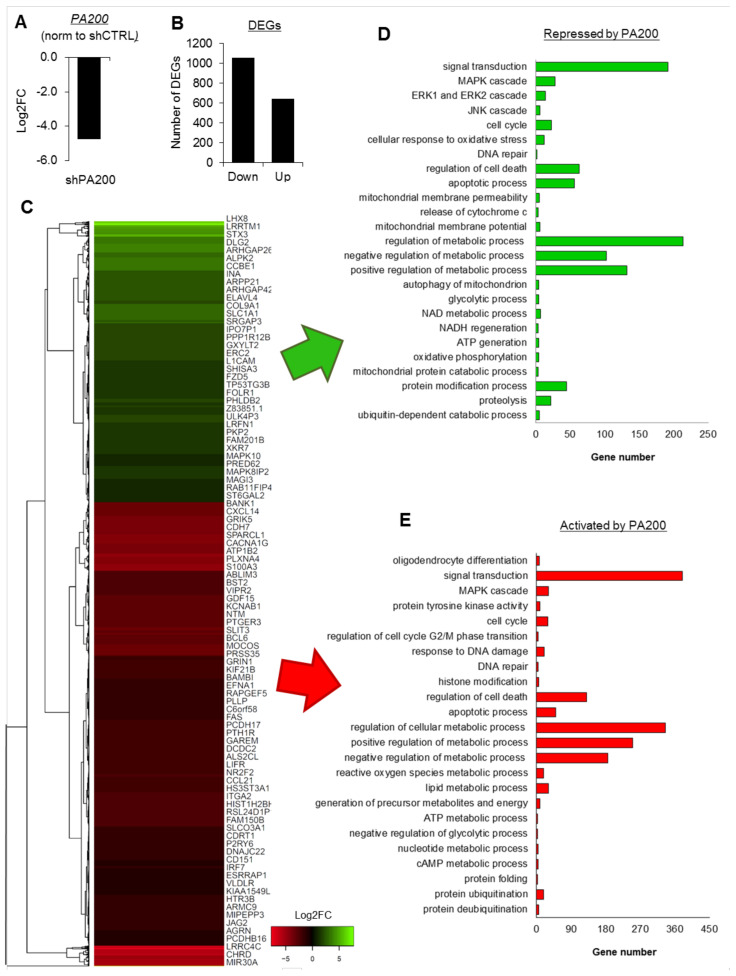
Deficiency of PA200 affects transcription of functionally relevant genes. (**A**) Analysis of RNA-Seq data confirmed depletion of *PSME4* (gene name for PA200) mRNA in shPA200 knockdowns in SH-SY5Y cells. (**B**) Quantification of differential gene expression (DEGs) between shCTRL and shPA200 disclosed the subset of genes up- and down-regulated in PA200-silenced cells. (**C**) The heat map shows Log2FC of DEGs, which were clustered according to Euclidean distance measurements. The list on the right quotes every second gene. (**D**,**E**) The GO enrichment analysis for the genes activated and repressed after PA200 silencing.

**Figure 2 ijms-22-01629-f002:**
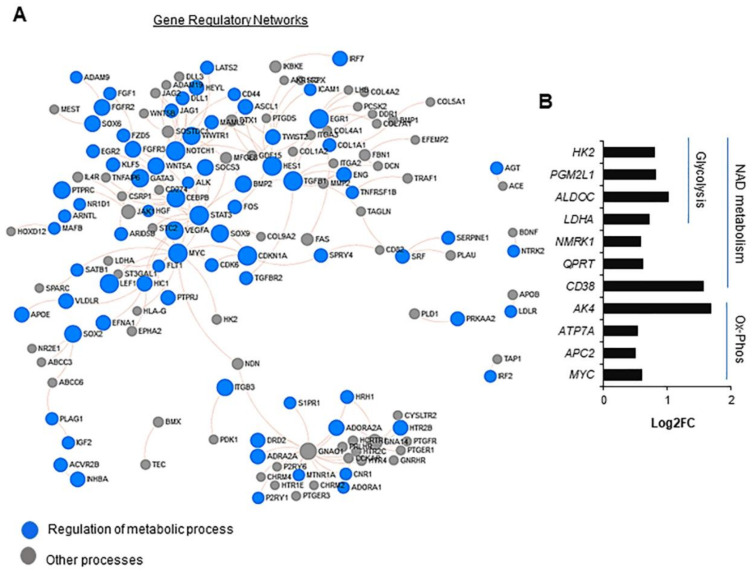
PA200 regulates the transcription of genes involved in cellular metabolism. (**A**) Examination of the minimal signaling network of genes transcriptionally affected by PA200 deficiency demonstrates the considerable representation of nodes, which are functionally linked to cellular metabolism. (**B**) PA200 silencing activated genes that contribute to NAD metabolism and mitochondria respiration. Differential gene expression is shown (shPA200 vs. shCTRL) derived from RNA-Seq data for NAD metabolic process (GO: 0019674), glycolytic process (GO: 0006096), and regulation of oxidative phosphorylation (GO: 0002082).

**Figure 3 ijms-22-01629-f003:**
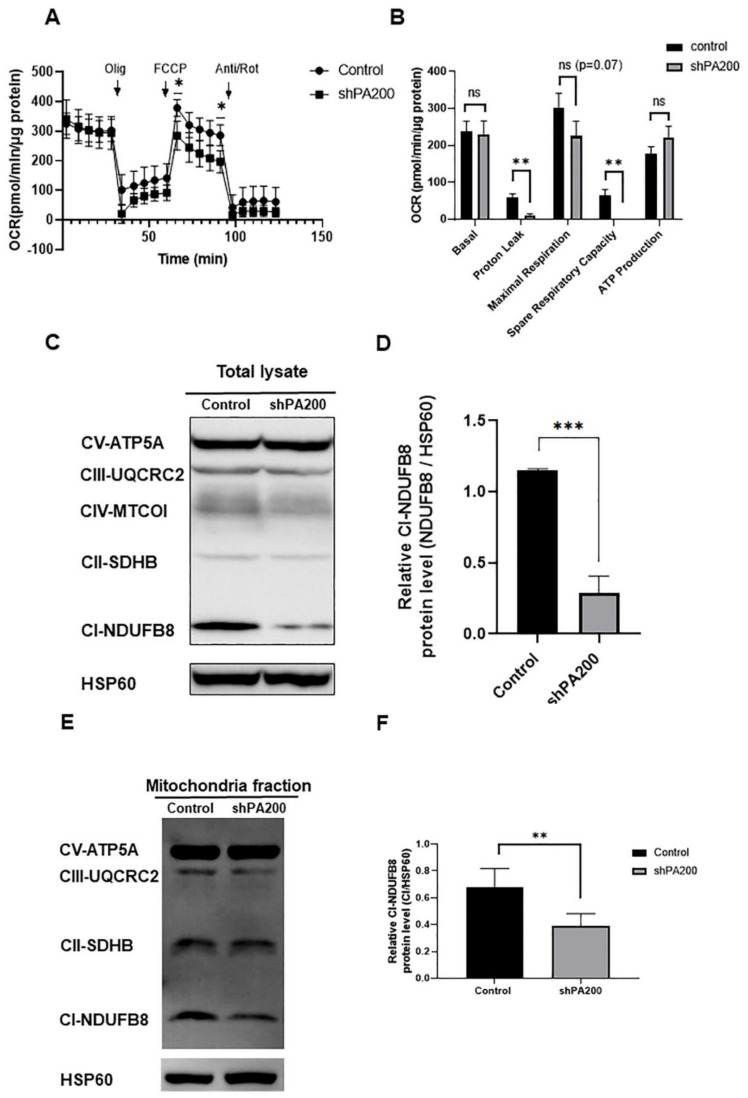
The genetic ablation of PA200 results in deficient mitochondrial function. (**A**) Oxygen consumption rate (OCR) was measured using the Seahorse XF 96 to analyze mitochondrial function. Control and shPA200 cells were seeded at a density of 35,000 cell/well in Seahorse XF 96 plates. The following day, the basal OCR was determined for 30 min after sequentially injecting mitochondrial drugs, including 1.5 µM oligomycin (Olig), 1 µM FCCP, and 1 µM of rotenone/antimycin-A cocktail (Anti/Rot). Data were analyzed using Wave Desktop software and presented as mean values ± SD (*n* = 3). Statistical analysis was performed using unpaired *t*-tests. (* indicates *p* < 0.05) (**B**) Calculated basal respiration, proton leak-linked respiration, maximal mitochondrial respiratory capacity, spare respiratory capacity, and ATP-coupled respiration. Data were normalized to total protein (pmol/min/µg protein). Data were analyzed using Wave Desktop software and presented as mean values ± SD (*n* = 3). Statistical analysis was performed using unpaired *t*-tests. (** indicates *p* < 0.01 and ns indicates not significant). (**C**) Total cell lysates from control and shPA200 cells were analyzed by western blot. OXPHOS antibody cocktail was used to detect four subunits of the electron transport chain (ETC) complexes, CI-NDUFB8, CII-SDHB, CIII-UQCRC2, and CIV-MTCOI, and one subunit for the ATP synthase, CV-ATP5A. Images were taken using a ChemiDoc Imager, and the density of the protein was normalized to the mitochondrial Hsp 60. (**D**) Statistical histogram of CI-NDUFB8 subunit from total lysates of PA200-deficient and control cells. Data were analyzed using Image Lab V 6.1 software and presented as mean values ± SD (*n* = 4). Groups were compared using unpaired *t*-tests. (*** indicates *p* < 0.001). (**E**) Mitochondria fractionation from control and shPA200 cells were performed and analyzed by western blot. The OXPHOS antibody cocktail was used to detect four subunits of the electron transport chain (ETC) complexes, CI-NDUFB8, CII-SDHB, CIII-UQCRC2, CIV-MTCOI, and one subunit for the ATP synthase, CV-ATP5A. Images were taken using a ChemiDoc Imager and the density of the protein was normalized to the mitochondrial Hsp 60. (**F**) Statistical analysis of the CI-NDUFB8 subunit from mitochondrial sufractions of shPA200 and control cells. Data were analyzed using Image Lab V 6.1 software and presented as mean values ± SD (*n* = 4). Groups were compared using unpaired *t*-tests. (** indicates *p* < 0.01).

**Figure 4 ijms-22-01629-f004:**
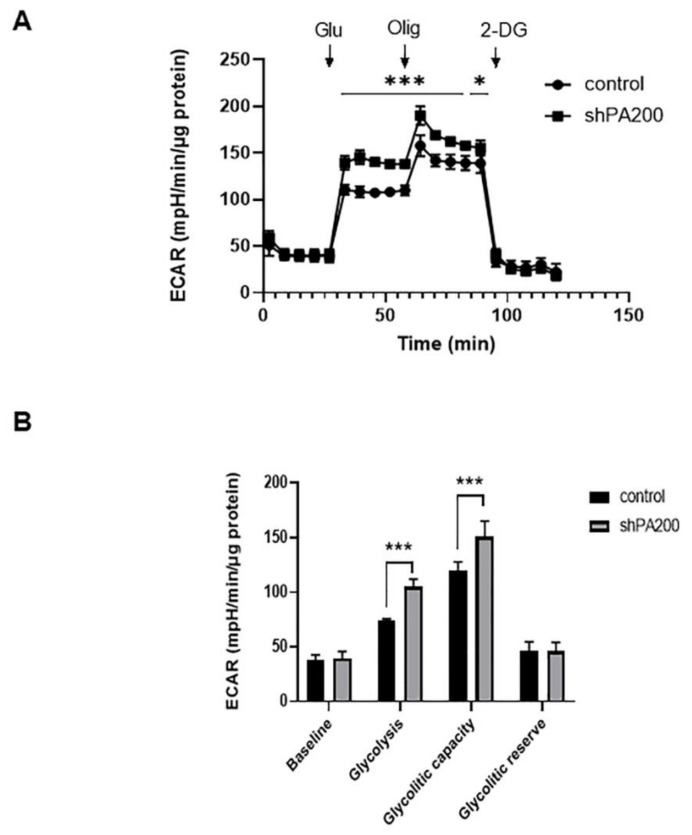
Increased glycolysis and glycolytic capacity in PA200 knockdown cells. (**A**) Glycolytic profile of PA200-depleted and control cells. Glycolysis was measured using the Seahorse XF 96 analyzer and was assessed by measuring the extracellular acidification rate (ECAR). The shPA200 and control cells were seeded in an XF-96 cell-culture microplate at 35,000 cells/well for 24 h. The following day, the media was replaced with glucose-free XF assay medium, and cells were incubated for 1 h without CO_2_. The basal ECAR was determined for 30 min prior to the injection of 10 mM glucose (Glu), 1 µM oligomycin (Olig), and 50 mM 2-deoxy-d-glucose (2-DG). (**B**) Calculated values of glycolysis and glycolytic capacity of shPA200 and control cells normalized to total protein (mpH/min/µg protein). Data were analyzed using Wave Desktop software. Data are presented as mean values ± SD (*n* = 4), and groups were compared using unpaired *t*-tests. (* indicates *p* < 0.05 and *** indicates *p* < 0.001).

**Figure 5 ijms-22-01629-f005:**
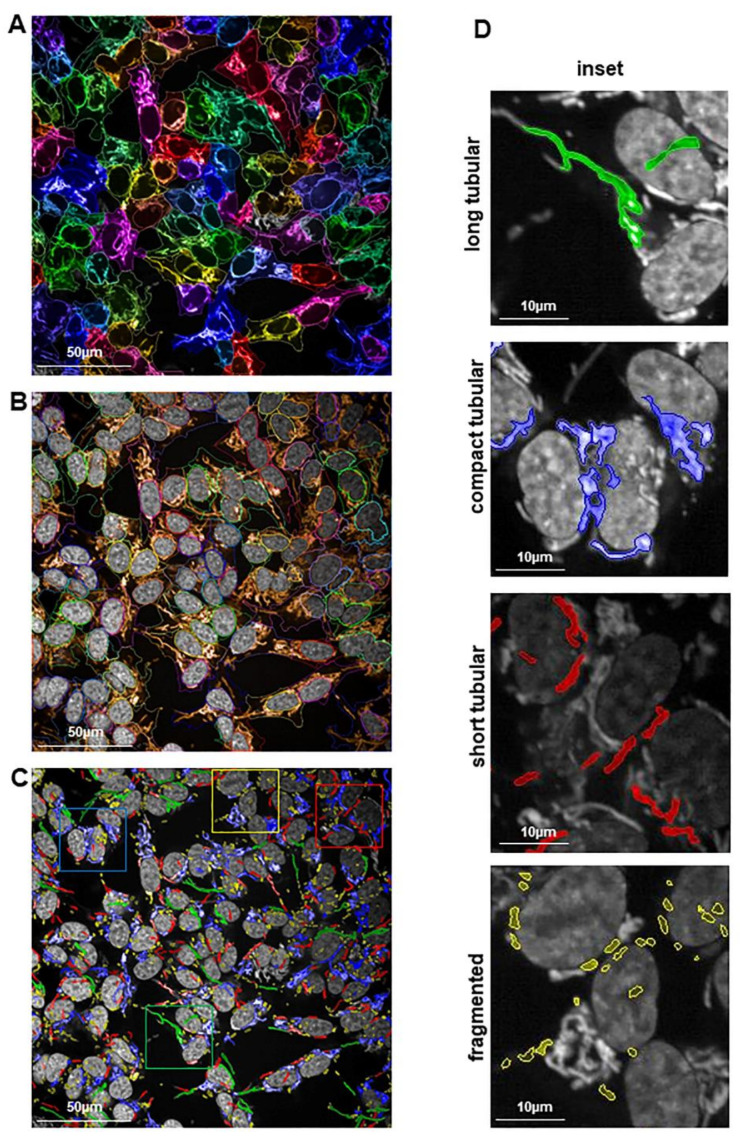
Classification of mitochondrial morphology using a high-content-screening (HCS) system. (**A**–**C**) Representative confocal images for mitochondria classification setting in live cells stained with MitoTracker Red CMXRos (mitochondria labeling), Hoechst (nuclei staining), and eGFP (cytoplasm). Both control and shPA200 are stably expressing the GFP (scale bar = 10 µm). Automated confocal microscopy was performed on an Opera Phenix high-content-screening system (Perkin Elmer). Image-acquisition settings were as follows: 63× water objective (NA = 1.15), appropriate lasers, and filters for Hoechst, eGFP, and Mitotracker Red in sequential mode to exclude spectra overlap. (**D**) Classification of mitochondria species was performed with the built-in Harmony 4.8 and PhenoLogic machine-learning software. Mitochondria were classified as long tubular area, small tubular area, compacted area, and fragmented area.

**Figure 6 ijms-22-01629-f006:**
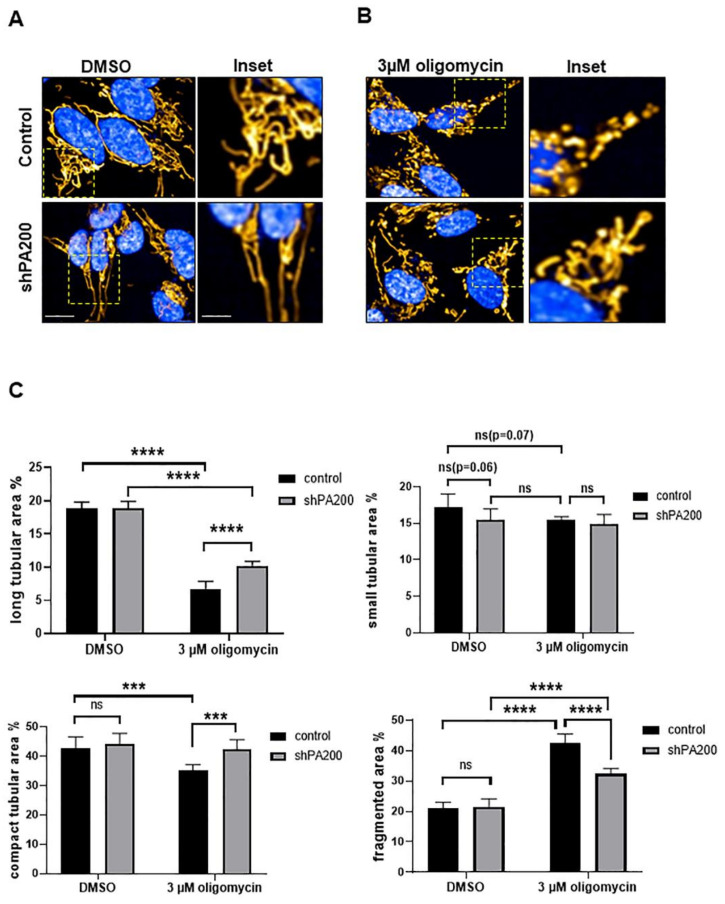
Reduced mitochondrial fragmentation in shPA200 cells after selective mitochondrial inhibitor treatment. Automated confocal microscopy was performed on an Opera Phenix high-content-screening system. Image-acquisition settings were as follows: 63× water objective (NA = 1.15), appropriate lasers and filters for Hoechst, EGFP, and Mitotracker Red CMXRos in sequential mode. (**A**,**B**) Representative confocal images of shPA200 and control cells stained with MitoTracker Red CMXRos after treatment with vehicle (DMSO) or 3 µM oligomycin for 24 h (scale bar = 10 µm, inset 2.5 µm). (**C**) Quantification of mitochondrial classes was performed with the built-in Harmony 4.8 and PhenoLogic machine-learning software. Mitochondria were classified as long tubular area, small tubular area, compacted area, and fragmented area. Data are presented as the mean ± SD of four biological replicates. Statistical analysis was performed by two-way ANOVA (*** indicates *p* < 0.001, **** indicates *p* < 0.0001 and ns indicates not significant).

**Figure 7 ijms-22-01629-f007:**
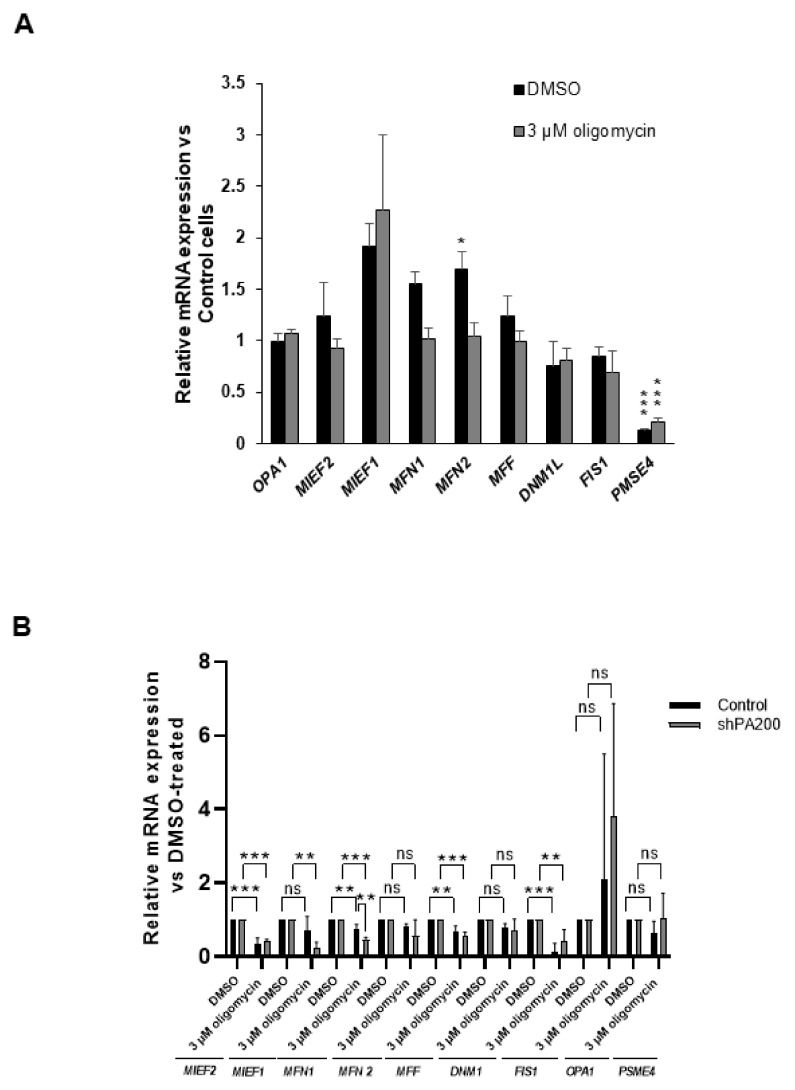
The genetic ablation of PA200 does not alter the gene expression of mitochondrial fission and fusion proteins compared to control, but oligomycin treatment does. The mRNA levels of mitochondrial fission and fusion genes were analyzed using quantitative real-time PCR analyses. Before RNA extraction, shPA200 and control cells were incubated with vehicle (DMSO) or the mitochondrial inhibitor, 3 µM oligomycin, for 24 h. (**A**) The mRNA levels of shPA200 were normalized to control cells for the respective treatment. (**B**) The mRNA expression in control cells and shPA200 cells treated with 3 µM oligomycin were normalized to the respective DMSO-treated cells. Puromycin was removed 24 h before performing every experiment. Data are presented as the mean ± SD of four separate experiments. Groups were compared using ANOVA. (* indicates *p* < 0.05, ** indicates *p* < 0.01, *** indicates *p* < 0.001 and ns indicates not significant).

**Figure 8 ijms-22-01629-f008:**
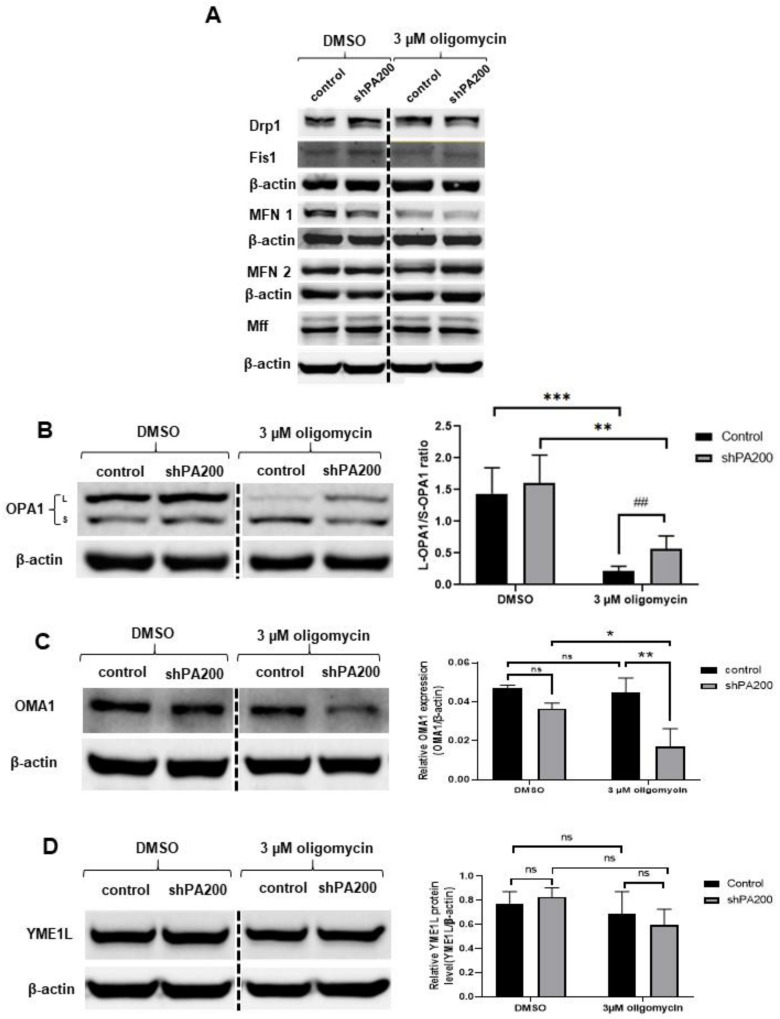
PA200-deficient cells exhibit low stress-induced L-OPA1 cleavage. Control and PA200-depleted cells were treated with either vehicle (DMSO) or 3 µM oligomycin for 24 h. Cells were lysed with RIPA buffer and an equal amount of protein was separated using SDS-PAGE and analyzed by western blot. (**A**) The protein levels of mitochondrial fission and fusion proteins were examined using anti-Drp1, anti-Fis1, anti-Mfn1, anti-Mfn2, and anti-Mff antibodies. β-actin was used as an internal loading control. (**B**) Mitochondrial fusion protein levels were examined using an anti-Opa1 antibody. Representative images of western blots and statistical analysis of the relative protein levels are shown. Images were taken using a ChemiDoc imager, and the pixel intensity was quantified and normalized to the internal control, β-actin. The ratio of L-Opa1 to S-Opa1 was determined. Data are presented as mean values ± SD (*n* = 4). Groups were compared using two-way ANOVA and *t*-tests. (** indicates *p* < 0.01, *** indicates *p* < 0.001, ## indicates *p* < 0.01) (**C**) IM peptidase OMA1 protein levels were examined using an anti-OMA1 antibody. Representative images of western blot and statistical analysis of the relative protein level are shown. Images were taken using a ChemiDoc imager, and the pixel intensity was quantified and normalized to the internal control, β-actin. Data are presented as mean values ± SD (*n* = 4). Statistical analysis was performed using two-way ANOVA. (* indicates *p* < 0.05, ** indicates *p* < 0.01 and ns indicates not significant). (**D**) Levels of the *i*-AAA protease, YME1L, were examined using an anti-YME1L antibody. Representative images of western blots and statistical analysis of the relative protein level are shown. Images were taken using a ChemiDoc imager, and the pixel intensity was quantified and normalized to the internal control, β-actin. Data are presented as mean values ± SD (*n* = 4). Groups were compared using two-way ANOVA. (ns indicates not significant).

**Table 1 ijms-22-01629-t001:** Primers used in this study.

Gene Name	Forward Primer (5′–3′)	Reverse Primer (5′–3′)
hPSME4	ATGGAGAGTGCCTGAACTATTG	GTAGGTCAGCACACTTCCTATTC
hFIS1	AGCTGGTGTCTGTGGAGGAC	ACGATGCCTTTACGGATGTC
hMFN1	CGGAACTTGATCGAATAGCC	AGAGCTCTTCCCACTGCTTG
hMFN2	ATGCATCCCCACTTAAGCAC	AGCACCTCACTGATGCCTCT
hDNM1L	AGATCTCATCCCGCTGGTC	CAGATCCTCGAGGCAAGAAG
hMIEF2	GCAGAGTTCTCCCAGAAACG	GTCTGCCTTGGTGTCATCCT
hMIEF1	GCAAAGGCAAGAAGGATGAC	CTTCATGTCCCTGTTCAGCA
hOPA1	CACTTCCTGGGTCATTCCTG	TGCTTCGTGAAACCAGATGT
hMFF	AAACGCTGACCTGGAACAAG	TTTTCAGTGCCAGGGGTTTA
hβ-actin	GACCCAGATCATGTTTGAGACC	CATCACGATGCCAGTGGTAC

**Table 2 ijms-22-01629-t002:** Primary antibodies used in this study.

Antibody	Source	Catalog Number	Host	Dilution
Drp1	BD Biosciences	# 611112	Mouse	1:1000
Mfn1	Abnova	# H00055669-M04	Mouse	1:1000
Mfn2	Sigma Aldrich	# WH0009927M3	Mouse	1:800
Opa1	Novus Biologicals	# NB110-55290	Rabbit	1:1000
OXPHOS	Abcam	# ab110413	Mouse	1:250
OMA1	SantaCruz Biotechnology	# sc-515788	Mouse	1:500
YME1L	Proteintech	# 11510-1-AP	Rabbit	1:1000
Fis1	Invitrogen, Thermo Fisher	# PA1-41082	Rabbit	1:1000
Hsp60	Invitrogen, Thermo Fisher	# MA3-012	Mouse	1:1000
β-actin-HRP	SantaCruz Biotechnology	# sc-1616	Goat	1:5000

## Data Availability

The data that support the findings of this study are available from the corresponding author upon reasonable request.

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
