# Peer review of "Cells Lacking PA200 Adapt to Mitochondrial Dysfunction by Enhancing Glycolysis via Distinct Opa1 Processing"

_ijms, 2021, doi:10.3390/ijms22041629_

Round 1

Reviewer 1 Report

In this study Douida et al. report that cells lacking PA200 adapt to mitochondrial dysfunction by enhancing glycolysis via distinct OPA1 processing. The reviewer thinks that the article in its current form is not acceptable for publication for several reasons. The main issue is, most of the results are not convincing and it is highly doubtful that the "mitochondrial" phenotype is of any biological significance. The major concern is that the observed phenotype is caused by apoptosis.

In Figure 1 the authors show that the main affected pathways are cell death and apoptosis. They did not discuss or analyse these results but concentrated on the OXPHOS System, which seems to to be just minorily affected.

Caspases indirectly regulate cleavage of the mitochondrial fusion GTPase OPA1 in neurons undergoing apoptosis (PMID: 19046944). OPA1 processing in cell death and disease - the long and short of it (PMID: 27189080).

Furthermore, some articles showed that several subunits of OXPHOS complexes especially complex I are cleaved during apoptotic processes e.g "Disruption of Mitochondrial Function during Apoptosis Is Mediated by Caspase Cleavage of the p75 Subunit of Complex I of the Electron Transport Chain (PMID: 15186778)".

The authors write that they did statistics for Fig 3 D with an n=4. They should show the whole blots not just two lanes. Fig.3 D, Fig. 3 E, Fig.8 A,B,C,D  

Minor Point 1: No one familiar with mitochondrial metabolism, would state that MYC and APC2 are "OXPHOS Proteins". 

Author Response

Authors’s response to reviewers

Thank you for giving us the opportunity to resubmit a revised version of the manuscript “Cells lacking PA200 adapt to mitochondrial dysfunction by enhancing glycolysis via distinct Opa1 processing” for publication in IJMS. We appreciate the time and effort that the reviewers dedicated to provide feedback on our manuscript and are grateful for the comments and suggestions to improve our manuscript.

According to the suggestions, we have thoroughly revised our manuscript and the final version is enclosed. The changes are highlighted within the manuscript. Point-by-point responses to the comments are listed below.

Reviewer 1:

Comment 1

„In this study Douida et al. report that cells lacking PA200 adapt to mitochondrial dysfunction by enhancing glycolysis via distinct OPA1 processing. The reviewer thinks that the article in its current form is not acceptable for publication for several reasons. The main issue is, most of the results are not convincing and it is highly doubtful that the "mitochondrial" phenotype is of any biological significance. The major concern is that the observed phenotype is caused by apoptosis.”

In Figure 1 the authors show that the main affected pathways are cell death and apoptosis. They did not discuss or analyse these results but concentrated on the OXPHOS System, which seems to to be just minorily affected.

Response 1:

We would like to thank the reviewer’s comment. While we appreciate the reviewer’s feedback, we respectfully disagree.

In our previously published manuscript “The proteasome activator PA200 regulates expression of genes involved in cell survival upon selective mitochondrial inhibition in neuroblastoma cells”by Douida, Batista et al., (PMID: 32368861) we observed that silencing PA200 did not affect cellular viability in vehicle treated samples compared to vehicle treated control cells. We also demonstrated that the loss of PA200 did sensitize cells to rotenone, but not to oligomycin-induced cell death. We showed by sulphorhodamine B (SRB) assay that cell viability in shPA200 cells was not significantly different compared to control cells after 3 µM oligomycin treatment. Necrosis of shPA200 cells was also similar to control cells after treatment with oligomycin measured by LDH release. Furthermore, we measured apoptotic markers including PARP1, phosphorylated γH2AX histone and we showed that oligomycin treatment did not affect PARP1 cleavage. Phosphorylated γH2AX histone level was slightly increased in both control and shPA200 cells. Cell death of PA200 depleted cells were comparable to control cells upon oligomycin treatment; however, our cell cycle analysis demonstrated a significantly increased cell population in the S phase upon PA200 depletion suggesting maintained cell proliferation with a possible DNA replication delay. Thus, we believe that the observed phenotype upon oligomycin treatment is not due to apoptosis in shPA200 cells. To the contrary we speculate that shPA200 try to escape from cell death by shifting metabolism to glycolysis.

In the published manuscript we evaluated the effect of PA200 depletion by analyzing the expression pattern of selected genes related to cell cycle progression, apoptosis and mitochondrial function and we concluded that depletion of PA200 selectively sensitize cells to different mitochondrial inhibitors, and the response was associated with transcriptional changes according to our ChIP-seq and mRNA expression data analysis.

Furthermore, our preliminary data and previously published manuscripts showed the involvement of Blm10 –the yeast ortholog of PA200-in mitochondrial phenotype (PMID: 24604417, ), and many studies discuss the link between proteasome activity and mitochondrial function in many disease, thus we were tempted to further explore the role of PA200 in a human cellular model.

We do agree that the GO presentation in Fig.1 D and Fig. 1E can be misleading because we presented intracellular processes (cell death, apoptosis) with very high GOs. Thus, for metabolism-relevant aspects Figure1 showed lower number of genes assigned to particular GOs. This is because we intended to emphasize that PA200 deficiency affects various facets of metabolic processes.

Prompted by the Reviewer’s comment we decided to extend Fig.1D and 1E, and have included the main GO terms such as regulation of metabolic process, signal transduction and regulation of cell death, which span narrower GO and more specific terms in Fig. 1D and 1E. Such an approach will strikingly represent the role of PA200 particularly in cellular metabolism. We replaced Figure 1 with a corrected, extended, modified new Figure 1.

Comment 2:

„Caspases indirectly regulate cleavage of the mitochondrial fusion GTPase OPA1 in neurons undergoing apoptosis (PMID: 19046944). OPA1 processing in cell death and disease - the long and short of it (PMID: 27189080).

Furthermore, some articles showed that several subunits of OXPHOS complexes especially complex I are cleaved during apoptotic processes e.g "Disruption of Mitochondrial Function during Apoptosis Is Mediated by Caspase Cleavage of the p75 Subunit of Complex I of the Electron Transport Chain (PMID: 15186778)".

Responses 2:

We appreciate the suggested scientific literatures from the Reviewer. In the initial manuscript, we included the manuscript by MacVicar and Langer (PMID: 27089080). In addition we also included another manuscript by MacVicar et al.. These manuscripts demonstrate that L-OPA1 and the accumulation of L-OPA1 are involved in mitochondrial fusion and protects cells from apoptosis. Oligomycin treatment leads to OPA1 proteolytic processing resulting in the accumulation of S-OPA1. Many studies also link uncleaved L-OPA1 to mitochondrial fusion and cleaved S-OPA1 to mitochondrial fission. The results of Figure 6 lead us to hypothesize that the significantly elevated compact (fused) mitochondria species are the results of different OPA1 processing of shPA200 cells compared to control. Results Figure 8 confirmed that oligomycin-induced L-OPA1 cleavage was attenuated in shPA200 cells. Thus, in our discussion we concluded that mitochondrial fragmentation is induced by oligomycin and that the fragmentation is related to the accumulation of S-OPA1 in control cells in accordance with previous studies. In PA200-deficient cells, the reduced fragmentation of mitochondria might originate from the accumulation of uncleaved L-OPA1 due to reduced OMA1. Our finding also indicates that the enhanced glycolysis might originate from the significantly reduced mitochondrial fragmentation via L-OPA1 accumulation. The significantly increased glycolysis might provide an alternative route from apoptosis in cells lacking PA200.

The manuscript (PMID: 19046944) referred by the reviewer describes that OPA1 processing might be regulated indirectly by caspases. The model the authors propose is that OPA1 is cleaved by an intermembrane protease (the authors do not mention which one(s)). The authors suggest that this protease is regulated by caspases. We would like to thank the reviewer for the suggestion. We run some preliminary experiments and we have not seen significant differences in the level of caspase-3 and intracellular cytochrome c (data not shown). We think that this direction based on our current data is too preliminary and would need more experiments, which would go beyond the scope of this study.

Comment 3:

„The authors write that they did statistics for Fig 3 D with an n=4. They should show the whole blots not just two lanes. Fig.3 D, Fig. 3 E, Fig.8 A,B,C,D „

Response 3:

We performed statistical analysis of each experiment of the manuscript. We included the original, whole, uncropped, uncut Western blots (each biological replicate) of Figure 3 in Supplementary Figure 4, and we included the original western blots of Figure 8 in Supplementary Figures 2 and 3.

Comment 4:

“Minor Point 1: No one familiar with mitochondrial metabolism, would state that MYC and APC2 are "OXPHOS Proteins".

Response 4:

We appreciate the comment. Indeed, MYC and APC2 are not directly involved in oxidative phosphorylation. However, these two candidates do belong to the GO Term: 0002082 – Regulation of oxidative phosphorylation. The authors never stated that MYC and APC2 are OXPHOS proteins in the submitted manuscript. We performed RNA-seq followed by whole transcriptome analysis and we identified significant gene expression changes of MYC and APC2. As indicated in the figure legend, the chart in Fig. 2B shows DEGs (shPA200 versus shCTRL) that were assigned to the three GO terms, which are of our particular interest in the submitted paper. The functional annotation of DEGs was carried out in Panther using statistical overrepresentation test. We also checked manually GO:0002082 in AmiGO2 and found both MYC and APC2 in the list of genes assigned to this GO term (http://amigo.geneontology.org/amigo/term/GO:0002082).

Regardless of the result of bioinformatics analysis, there are numerous reports, which link MYC and APC2 with the mitochondrial activity and, therefore, possibly with regulation of oxidative phosphorylation. For example, APC2 – “calcium-dependent mitochondrial carrier protein that catalyses the import of ATP co-transported with metal divalent cations across the mitochondrial inner membrane in exchange for phosphate (Pi) (PubMed:22062157, PubMed:28695448, PubMed:26140942, PubMed:26444389).

As for MYC, the protein was shown to regulate mitochondrial biogenesis, respiration and also oxidative phosphorylation (PMID: 20955683, PMID: 28978427). Furthermore, by functionally interacting with transcription factors such as E2F1 or HIF1; c-MYC controls glucose and glutamine metabolism by inducing genes involved in nucleotide metabolism, microRNAs that homeostatically attenuate E2F1 expression, induces a transcriptional program for hypoxic adaptation, regulates expression of glycolytic genes including lactate dehydrogenase A (LDHA), represses miR-23a/b to increase glutaminase (GLS) protein expression and glutamine metabolism. In cancers, MYC was postulated to concurrently drive aerobic glycolysis and/or oxidative phosphorylation to provide sufficient energy and anabolic substrates for cell growth and proliferation (PMID: 19861459). The role of MYC in coordination between glycolysis and OXPHOS in well-oxygenated environments has been recently reviewed in PMID: 29706933.

Reviewer 2 Report

In this manuscript authors found the effects of loss of proteosomal activator gene PA200. From RNA-seq data they found their role in mitochondrial oxidative phosphorylation activity. They have shown through various experiments that depletion of PA200 induced mitochondrial dysfunction and increased glycolysis. They have also shown that loss of PA200 also affects OPA-1 processing. Overall manuscripts is well written and all the results articulated well with cohesive discussion. Before acceptance of this manuscript, I would suggest including some experiments in this manuscript.

  1. Measure the activity of LDH and Pyruvate dehydrogenase to show more glycolysis leading to anaerobic respiration.
  2. Measure the activity of all the complex of OXPHOS.
  3. What is the physiological and pathological implications of shift from OXPHOS to glycolysis, Do they any role in cancer because supporting the idea of Warburg effects.

Author Response

Authors’s response to reviewers

Thank you for giving us the opportunity to resubmit a revised version of the manuscript “Cells lacking PA200 adapt to mitochondrial dysfunction by enhancing glycolysis via distinct Opa1 processing” for publication in IJMS. We appreciate the time and effort that the reviewers dedicated to provide feedback on our manuscript and are grateful for the comments and suggestions to improve our manuscript.

According to the suggestions, we have thoroughly revised our manuscript and the final version is enclosed. The changes are highlighted within the manuscript. Point-by-point responses to the comments are listed below.

Reviewer 2:

Comment 1

“In this manuscript authors found the effects of loss of proteosomal activator gene PA200. From RNA-seq data they found their role in mitochondrial oxidative phosphorylation activity. They have shown through various experiments that depletion of PA200 induced mitochondrial dysfunction and increased glycolysis. They have also shown that loss of PA200 also affects OPA-1 processing. Overall manuscripts is well written and all the results articulated well with cohesive discussion. Before acceptance of this manuscript, I would suggest including some experiments in this manuscript.”

Response 1

We highly appreciate the reviewer’s comment. Our point-by point responses are found below:

Comment 2:

„Measure the activity of LDH and Pyruvate dehydrogenase to show more glycolysis leading to anaerobic respiration.”

„Measure the activity of all the complex of OXPHOS.”

Response 2:

We appreciate the reviewer’s suggestion, however we would like to state the following:

We used the technology of Agilent Seahorse to conduct our experiments to study mitochondrial function. The technology measures energy metabolism in live cells in real time and the results provide information directly related to cellular function. In our manuscript, we focused to measure cellular bioenergetics and we assessed cellular respiration and glycolysis to obtain a metabolic description of our cell model.

Conventional assays including enzyme activity measurements are end-points measurements and often result in a static view of metabolism. We were able to measure with Agilent Seahorse technology the kinetic activity of mitochondrial respiration and glycolysis. Mitochondrial activity is quantitatively measured by the oxygen consumption rate (OCR). Glycolysis is measured quantitatively by the extracellular acidification rate (ECAR). The values we show in Figures 3B and 4B are calculated from the OCR measurements for the mitochondrial activity and from the ECAR measurements for the glycolytic activity using Wave 2.3 Agilent Seahorse desktop software. These assays are standardized and commonly used in research represented by more than 5000 peer-reviewed manuscripts (PMID: 30725464, 29197011, 31275241, 31917509).

Comment 3:

„What is the physiological and pathological implications of shift from OXPHOS to glycolysis, Do they any role in cancer because supporting the idea of Warburg effects.”

Response 3:

We appreciate the comment. The role of PA200 in cancer or in tumorigenesis has been described so far in one published manuscript. In this manuscript the authors showed that miR29b targets the gene of PA200 (PSME4) and enhanced the antimyeloma activities of bortezomib, an anticancer drug and a proteasome inhibitor. The hallmark of metabolic reprogramming, accelerated glucose consumption or low oxidative phosphorylation in cancers cells describe the phenomena called Warburg effect. Aerobic glycolysis provide cancers cells the abilities to meet the energy demand to rapidly proliferate or survive. It is possible that PA200 might be involved in the regulation of such process via proteasomal degradation or transcriptional regulation. The results of our manuscript might point to such direction, however to explore such idea would need more experiments, which would go beyond the scope of this study.

In our previously published manuscript (PMID: 32368861) we showed no significant cell death and an accumulation of shPA200 cells in the S phase after oligomycin treatment compared to control. This might be indicative of slower cell growth during the switch from OXPHOS to inefficient glycolysis in PA200 deficient cells. Taken together we can speculate that the mitochondrial dysfunction we observed in PA200 deficient cells and the suppression of OXPHOS system might result in some level of Warburg effect.

Reviewer 3 Report

In this Manuscript the authors demonstrate a possible role of the proteosome activator PA200 depletion (using shPA200) on the overall transcriptome of SH-SY5Y cell, as well as on cell metabolic activity, notably mitochondrial function, with a decrease in OXPHOS and increase in glycolysis (although these effects are very small), also with effects in terms of mitochondrial dynamics. Overall this is a well-thought study, carried out with appropriate methodology. Thereare a few issues that the authors should consider in a revised version.

1- More definite proof of the specific and stable deletion of PA200 depletion  in the shRNA cell line should be should (i.e. Western Blot). This could be supplemental. I appreciate that the authors may have this from previous work, but it is important for the consideration of this manuscript.

2- All the Western blots throughout should be shown in full as Supplementary data.

3- The differences in OCR and ECAR are VERY small, especially the former. The authors should be more forward with their data noting (in the abstyact and rest of the manuscript) that these are really minor changes. Moreover what the "stress" is should also be available in the Abstract. The specific coice of oligomycin as a stressor should also be expanded upon, This is a specific mitochondrial ATP-syntase inhibitor, and not usually employed at this level (unlike oxidative stress-rendering molecules, for example).

4- Some measure of mitochondrial activity (ie. JC1 and flow cytometry) should be performed. It is also odd why the authors did not monitor ROS levels, which have been linked to the types of phenomena described here.

Author Response

Authors’s response to reviewers

Thank you for giving us the opportunity to resubmit a revised version of the manuscript “Cells lacking PA200 adapt to mitochondrial dysfunction by enhancing glycolysis via distinct Opa1 processing” for publication in IJMS. We appreciate the time and effort that the reviewers dedicated to provide feedback on our manuscript and are grateful for the comments and suggestions to improve our manuscript.

According to the suggestions, we have thoroughly revised our manuscript and the final version is enclosed. The changes are highlighted within the manuscript. Point-by-point responses to the comments are listed below.

Reviewer 3:

Comment 1:

„In this Manuscript the authors demonstrate a possible role of the proteosome activator PA200 depletion (using shPA200) on the overall transcriptome of SH-SY5Y cell, as well as on cell metabolic activity, notably mitochondrial function, with a decrease in OXPHOS and increase in glycolysis (although these effects are very small), also with effects in terms of mitochondrial dynamics. Overall this is a well-thought study, carried out with appropriate methodology. Thereare a few issues that the authors should consider in a revised version.”

“More definite proof of the specific and stable deletion of PA200 depletion in the shRNA cell line should be should (i.e. Western Blot). This could be supplemental. I appreciate that the authors may have this from previous work, but it is important for the consideration of this manuscript.”

Response 1:

We highly appreciate the reviewer’s comments and suggestions. Following the Reviewer’s suggestion we included the Western-blot and the quantitative real-time PCR results of the stable depletion of PA200 as Supplementary Figure 1. We periodically confirmed the stable depletion of PA200 through our study. In the supplementary we included a western blot of PA200 depletion before we started our experiments and we also included a western blot of two independent experiments when cells were treated with vehicle and oligomycin during the study.

Comment 2:

„All the Western blots throughout should be shown in full as Supplementary data.”

Response 2

We provided the original western blots as Supplementary Figures 2, 4 and 5.

Comment 3:

„The differences in OCR and ECAR are VERY small, especially the former. The authors should be more forward with their data noting (in the abstyact and rest of the manuscript) that these are really minor changes. Moreover what the "stress" is should also be available in the Abstract. The specific coice of oligomycin as a stressor should also be expanded upon, This is a specific mitochondrial ATP-syntase inhibitor, and not usually employed at this level (unlike oxidative stress-rendering molecules, for example).”

Response 3:

We thank the Reviewer’s comment. While we appreciate the comment, we respectfully disagree. In our opinion, the differences are not very small; moreover, the differences are statistically significant. In Figure 3B and Figure 4B we calculated different values from the OCR measurements for the mitochondrial activity and from the ECAR measurements for the glycolytic activity, respectively. As Figure 3B demonstrates proton leak which can be a sign of mitochondrial damage, and reversed respiratory capacity dropped almost to zero in cells lacking PA200. Furthermore, glycolysis increased ~35 % and glycolytic capacity ~ 40 % in shPA200 cells (Figure 4B).

According to the Reviewer’s suggestion, we modified the abstract and we specified the selective mitochondrial inhibitor. We modified the abstract to:

“We show that stable knock-down of PA200 induces mitochondrial dysfunction, increases glycolysis, suggesting a shift from oxidative phosphorylation to glycolysis especially when cells are exposed to oligomycin-induced stress. Furthermore, the increased glycolysis of shPA200 after inhibition of ATP synthase by oligomycin is observed in association with preserved long- and compact tubular mitochondrial morphology. The present study also demonstrates that the proteolytic cleavage of Opa1 is affected, and that the level of OMA1 is significantly reduced in shPA200 cells upon oligomycin-induced mitochondrial insult. Together, these findings suggest a role for PA200 in the regulation of metabolic changes in response to selective inhibition of ATP synthase.”

Following the Reviewer’s suggestion, we extended the text of the manuscript explaning why we applied oligomycin. We modified the test from:

“We challenged cells with oligomycin, a selective mitochondrial ATPase synthase inhibitor. Oligomycin induces mitochondrial fragmentation (46). Furthermore, we recently showed that loss of PA200 does not sensitize cells to death after 3 µM oligomycin treatment, in contrast to rotenone treatment. We also demonstrated that PA200 depletion causes cells to accumulate in the S phase of the cell cycle after treatment with oligomycin, indicative of possible DNA replication (35).”

To:

“We challenged cells with oligomycin, a selective mitochondrial ATPase synthase inhibitor. Oligomycin induces mitochondrial fragmentation (46). Furthermore, we recently showed that loss of PA200 does not sensitize cells to death after 3 µM oligomycin treatment, in contrast to rotenone treatment. We also demonstrated that PA200 depletion causes cells to accumulate in the S phase of the cell cycle after treatment with oligomycin, indicative of possible delay of DNA replication (35). Oligomycin was described as an oncogenic agent in several types of lung cancer cells causing increased cell invasion and migration (47). Oligomycin was validated as a relevant tool to study bioenergetics adaptation to OXPHOS suppression in many cancer cell lines (48). Moreover, our ChIP-seq data showed PA200 enriched regions in the genome of SH-SY5Y and that the status of binding or eviction of PA200 to/from specific promoters depends on the exposure to selective mitochondrial toxins including oligomycin. GO annotation revealed that many genes that were significantly enriched in PA200 contribute to the regulation of metabolism (35).

Thus, as a logical step we also determinded the effects of PA200 deficiency on mitochondrial morphology after oligomycin treatment.”

Comments 4:

„Some measure of mitochondrial activity (ie. JC1 and flow cytometry) should be performed. It is also odd why the authors did not monitor ROS levels, which have been linked to the types of phenomena described here.”

Response 4:

We thank the Reviewer’s suggestion. JC1 is a measure of mitochondrial membrane potential. We measured mitochondrial membrane potential to verify mitochondrial status using TMRE. Tetramethylrhodamine ethyl ester (TMRE) loads specifically into polarized mitochondria. We only did not want to include it in the body of the manuscript because in our opinion it deviated our focus. However, if the reviewer agrees we would include the figure as a Supplementary Figure 3A.

We performed mitochondrial membrane potential measurements after treatment with the mitochondrial inhibitor. To detect mitochondrial membrane potential by flow cytometry, we used TMRE following 24 hr treatment with DMSO and 3 µM oligomycin. When cells were treated with 3 µM oligomycin, the mitochondrial membrane potential was significantly increased in both cell lines control and shPA200 compared to oligomycin treated control. In summary, the loss of PA200 slightly potentiated the effect of oligomycin, causing increased hyperpolarization of mitochondrial membrane potential.

We also detected intracellular ROS and included in the manuscript as Supplementary Figure 3B.

As we added in the text, the level of ROS was increased in both control and shPA200 cells, but only shPA200 cells showed significant increase of intracellular ROS level after oligomycin treatment.

We modified the text from:

“To determine whether the changes in mitochondrial metabolic profiling alter the levels of core proteins of the OXPHOS machinery, we used total cell lysates (Figure 3C and Supplementary Figure 2A) and purified mitochondria (Figure 3E and Supplemen-tary Figure 2B). Loss of PA200 led to a significantly reduced protein level of the acces-sory subunit NDUFB8, a member of Complex I. It is required for the assembly of the functional complex, but may not be involved in the catalytic activity of Complex I(Figure 3D and F) (44).”

To:

“To determine whether the changes in mitochondrial metabolic profiling alter the levels of core proteins of the OXPHOS machinery, we used total cell lysates (Figure 3C and Supplementary Figure 2A) and purified mitochondria (Figure 3E and Supplemen-tary Figure 2B). Loss of PA200 led to a significantly reduced protein level of the acces-sory subunit NDUFB8, a member of Complex I. It is required for the assembly of the functional complex, but may not be involved in the catalytic activity of Complex I(Figure 3D and F) (44).

Subsequently, we performed mitochondrial membrane potential measurements. To detect mitochondrial membrane potential by flow cytometry, we used TMRE fol-lowing 24 hr treatment with DMSO and 3 µM oligomycin. When cells were treated with 3 µM oligomycin, the mitochondrial membrane potential was significantly in-creased in both control and shPA200 cell compared to vehicle treated control. In sum-mary, the loss of PA200 slightly potentiated the effect of oligomycin, compared to con-trol cells, causing increased hyperpolarization of mitochondrial membrane potential. The effect however did not reach significance (Supplementary Figure 3A). We com-pared intracellular ROS in shPA200 and control cells using carboxy-H2DCFDA with and without oligomycin treatment. We found significantly elevated cytosolic ROS in oligomycin-treated shPA200 cells compared to vehicle treated shPA200 cells, which indicates existing oxidative stress in our cell model (Supplementary Figure 3B).”

We used the technology of Agilent Seahorse to conduct our experiments to study mitochondrial function. The technology measures energy metabolism in live cells in real time and the results provide information directly related to cellular function. In our manuscript, we focused to measure cellular bioenergetics and we assessed cellular respiration and glycolysis to obtain a metabolic description of our cell model.

We were able to measure with Agilent Seahorse technology the kinetic activity of mitochondrial respiration and glycolysis. Mitochondrial activity is quantitatively measured by the oxygen consumption rate (OCR). Glycolysis is measured quantitatively by the extracellular acidification rate (ECAR). The values we show in Figures 3B and 4B are calculated from the OCR measurements for the mitochondrial activity and from the ECAR measurements for the glycolytic activity using Wave 2.3 Agilent Seahorse desktop software. These assays are standardized and commonly used in research represented by more than 5000 peer-reviewed manuscripts (PMID: 30725464, 29197011, 31275241, 31917509).

Round 2

Reviewer 1 Report

The article is acceptable in its current form.

Author Response

We thank the Reviewer's useful comments, suggestion and support.

Reviewer 2 Report

I am happy that authors have considered my request  and included the experiments as suggested. Now, this manuscript can be accepted for further processing.

Author Response

(The authors gave the same response as above.)

Reviewer 3 Report

The authors have adequately addressed most of my concerns, except for one, Comment 3 in this response. The differences mentioned are small, and this is not an opinion, as I also have a vast experience with these assays, so I continue to (respectfully) disagree with the authors.

That these differences are statistically significant is not a good answer. In fact, if they were NOT statistically significant differences they would not be considered differences at all, as I'm sure the authors recognize. I am not questioning the differences, I'm questioning their magnitude, which may be physiologically relevant, or not.

So I maintain that a mention of this in the abstract is necessary to not mislead readers, as it is very clear in the results.

Author Response

Reviewer’s comment:

„The authors have adequately addressed most of my concerns, except for one, Comment 3 in this response. The differences mentioned are small, and this is not an opinion, as I also have a vast experience with these assays, so I continue to (respectfully) disagree with the authors.

That these differences are statistically significant is not a good answer. In fact, if they were NOT statistically significant differences they would not be considered differences at all, as I'm sure the authors recognize. I am not questioning the differences, I'm questioning their magnitude, which may be physiologically relevant, or not.

So I maintain that a mention of this in the abstract is necessary to not mislead readers, as it is very clear in the results.”

Answers:

We appreciate the Reviewer’s comment. Following his/her suggestion we modified the abstract and we included the numerical values of changes for both mitochondrial and glycolytic activities. We hope that the modification will satisfy the Reviewer and that the text is not going to be misleading. We, in addition included the values in the body of the text.

We modified the abstract to:

Abstract: The conserved Blm10/PA200 proteins are proteasome activators. Previously, we identified PA200-enriched regions in the genome of SH-SY5Y neuroblastoma cells by chromatin immunoprecipitiation (ChIP) and ChIP-seq analysis. We also found that selective mitochondrial inhibitors induced PA200 redistribution in the genome. Collectively, our data indicated that PA200 regulates cellular homeostasis at the transcriptional level. In the present study, our aim is to investigate the impact of stable PA200 depletion (shPA200) on the overall transcriptome of SH-SY5Y cells. RNA-seq data analysis reveals that the genetic ablation of PA200 leads to overall changes in the transcriptional landscape of SHSY5Y neuroblastoma cells. PA200 activates and represses genes regulating metabolic processes, such as glycolysis and mitochondrial function. Using metabolic assays in live cells we show that stable knock-down of PA200 does not change basal respiration. Spare respiratory capacity and proton leak however are slightly, but significantly reduced in PA200 deficient cells by 99.834 % and 84.147 %, respectively compared to control. Glycolysis and glycolytic capacity show 42.186 % and 26.104 % increase in shPA200 cells, respectively compared to control. These data suggest a shift from oxidative phosphorylation to glycolysis especially when cells are exposed to oligomycin-induced stress. Furthermore, we observed a preserved long- and compact tubular mitochondrial morphology after inhibition of ATP synthase by oligomycin, which might be associated with the glycolytic change of shPA200 cells. The present study also demonstrates that the proteolytic cleavage of Opa1 is affected, and that the level of OMA1 is significantly reduced in shPA200 cells upon oligomycin-induced mitochondrial insult. Together, these findings suggest a role for PA200 in the regulation of metabolic changes in response to selective inhibition of ATP synthase in an in vitro cellular model.
